There are amendments to this paper

# The observation of vibrating pear-shapes in radon nuclei

P.A. Butler [1], L.P. Gaffney [1,2], P. Spagnoletti[3], J. Konki [2], M. Scheck [3], J.F. Smith[3], K. Abrahams[4], M. Bowry[5], J. Cederkäll[6], T. Chupp [7], G. de Angelis[8], H. De Witte[9], P.E. Garrett[10], A. Goldkuhle[11], C. Henrich[12], A. Illana [8], K. Johnston [2], D.T. Joss[1], J.M. Keatings [3], N.A. Kelly[3], M. Komorowska[13], T. Kröll [12], M. Lozano[2], B.S. Nara Singh[3], D. O'Donnell[3], J. Ojala [14,15], R.D. Page[1], L.G. Pedersen[16], C. Raison[17], P. Reiter[11], J.A. Rodriguez[2], D. Rosiak[11], S. Rothe [2], T.M. Shneidman [18], B. Siebeck[11], M. Seidlitz[11], J. Sinclair[3], M. Stryjczyk [9], P. Van Duppen[9], S. Vinals[19], V. Virtanen[14,15], N. Warr[11], K. Wrzosek-Lipska[13] & M. Zielinska[20]

There is a large body of evidence that atomic nuclei can undergo octupole distortion and assume the shape of a pear. This phenomenon is important for measurements of electric-dipole moments of atoms, which would indicate CP violation and hence probe physics beyond the Standard Model of particle physics. Isotopes of both radon and radium have been identified as candidates for such measurements. Here, we observed the low-lying quantum states in $^{224}$Rn and $^{226}$Rn by accelerating beams of these radioactive nuclei. We show that radon isotopes undergo octupole vibrations but do not possess static pear-shapes in their ground states. We conclude that radon atoms provide less favourable conditions for the enhancement of a measurable atomic electric-dipole moment.

[1] Oliver Lodge Laboratory, University of Liverpool, Liverpool L69 7ZE, UK. [2] CERN, Geneva 23 CH-1211, Switzerland. [3] School of Computing, Engineering and Physical Sciences, University of the West of Scotland, Paisley PA1 2BE, UK. [4] Department of Physics & Astronomy, University of the Western Cape, Private Bag X17, Bellville 7535, South Africa. [5] TRIUMF, Vancouver V6T 2A3 BC, Canada. [6] Physics Department, Lund University, Box 118, Lund SE-221 00, Sweden. [7] Department of Physics, University of Michigan, Ann Arbor 48104 MI, USA. [8] INFN Laboratori Nazionali di Legnaro, Legnaro 35020 PD, Italy. [9] Instituut voor Kern- en Stralingsfysica, KU Leuven, Leuven B-3001, Belgium. [10] Department of Physics, University of Guelph, Guelph N1G 2W1 Ontario, Canada. [11] Institute for Nuclear Physics, University of Cologne, Cologne 50937, Germany. [12] Institut für Kernphysik, Technische Universität Darmstadt, Darmstadt 64289, Germany. [13] Heavy Ion Laboratory, University of Warsaw, Warsaw PL-02-093, Poland. [14] Department of Physics, University of Jyvaskyla, P.O. Box 35, Jyvaskyla FIN-40014, Finland. [15] Helsinki Institute of Physics, P.O. Box 64, Helsinki FIN-00014, Finland. [16] Department of Physics, University of Oslo, P.O. Box 1048, Oslo N-0316, Norway. [17] Department of Physics, University of York, York YO10 5DD, UK. [18] JINR Dubna, Dubna 141980 Moscow Region, Russia. [19] Consejo Superior De Investigaciones Científicas, Madrid S 28040, Spain. [20] IRFU CEA, Université Paris-Saclay, Gif-sur-Yvette F-91191, France. Correspondence and requests for materials should be addressed to P.A.B. (email: peter.butler@liverpool.ac.uk)

It is well established by the observation of rotational bands that atomic nuclei can assume quadrupole deformation with axial and reflection symmetry, usually with the shape of a rugby ball. The distortion arises from long-range correlations between valence nucleons, which becomes favourable when the proton and/or neutron shells are partially filled. For certain values of proton and neutron number it is expected that additional correlations will cause the nucleus to also assume an octupole shape ('pear-shape') where it loses reflection symmetry in the intrinsic frame[1]. The fact that some nuclei can have pear-shapes has influenced the choice of atoms having nuclei with odd nucleon number $A$ ($=Z+N$) employed to search for permanent electric-dipole moments (EDMs). Any measurable moment will be amplified if the nucleus has octupole collectivity and further enhanced by static-octupole deformation. At present, experimental limits on EDMs, that would indicate charge-parity (CP) violation in fundamental processes where flavour is unchanged, have placed severe constraints on many extensions of the Standard Model. Recently, new candidate atomic species, such as radon and radium, have been proposed for EDM searches. For certain isotopes octupole effects are expected to enhance, by a factor 100–1000, the nuclear Schiff moment (the electric-dipole distribution weighted by radius squared) that induces the atomic EDM[2–4], thus improving the sensitivity of the measurement. There are two factors that contribute to the greater electrical polarizability that causes the enhancement: (i) the odd-$A$ nucleus assumes an octupole shape; (ii) an excited state lies close in energy to the ground state with the same angular momentum and intrinsic structure but opposite parity. Such parity doublets arise naturally if the deformation is static (permanent octupole deformation).

The observation of low-lying quantum states in many nuclei with even $Z$, $N$ having total angular momentum ('spin') and parity of $I^\pi = 3^-$ is indicative of their undergoing octupole vibrations about a reflection-symmetric shape. Further evidence is provided by the sizeable value of the electric octupole (E3) moment for the transition to the ground state, indicating collective behaviour of the nucleons. However, the number of observed cases where the correlations are strong enough to induce a static pear-shape is much smaller. Strong evidence for this type of deformation comes from the observation of a particular behaviour of the energy levels for the rotating quantum system and from an enhancement in the E3 moment[5]. So far there are only two cases, $^{224}$Ra[6] and $^{226}$Ra[7] for which both experimental signatures have been observed. The presence of a parity doublet of 55 keV at the ground state of $^{225}$Ra makes this nucleus therefore a good choice for EDM searches[8]. In contrast to the radium isotopes, much less is known about the behaviour of radon (Rn) nuclei proposed as candidates for atomic EDM searches on account of possible enhancement of their Schiff moments[9–17]. For this reason, different isotopes of radon have been listed in the literature, for example $^{221,223,225}$Rn[14], each having comparable half-lives and ground state properties. The most commonly chosen isotope for theoretical calculations[9,10] and the planning of experiments[11–13] is $^{223}$Rn.

In this work, we present data on the energy levels of heavy even-even Rn isotopes to determine whether parity doublets are likely to exist near the ground state of neighbouring odd-mass Rn nuclei. Direct observation of low-lying states in odd-$A$ Rn nuclei (for example following Coulomb excitation or β-decay from the astatine parent) is presently not possible, as it will require significant advances in the technology used to produce radioactive ions. We observe that $^{224,226}$Rn behave as octupole vibrators in which the octupole phonon is aligned to the rotational axis. We conclude that there are no isotopes of radon that have static-octupole deformation, so that any parity doublets in the odd-mass

neighbours will not be closely spaced in energy. This means that radon atoms will provide less favourable conditions for the enhancement of a measurable atomic electric-dipole moment.

## Results

**Measurement of the quantum structure of heavy radon isotopes.** In the experiments described here, $^{224}$Rn ($Z = 86$, $N = 138$) and $^{226}$Rn ($Z = 86$, $N = 140$) ions were produced by spallation in a thick thorium carbide target bombarded by ~$10^{13}$ protons s$^{-1}$ at 1.4 GeV from the CERN PS Booster. The ions were accelerated in HIE-ISOLDE to an energy of 5.08 MeV per nucleon and bombarded secondary targets of $^{120}$Sn. In order to verify the identification technique, another isotope of radon, $^{222}$Rn, was accelerated to 4.23 MeV/u. The γ-rays emitted following the excitation of the target and projectile nuclei were detected in Miniball[18], an array of 24 high-purity germanium detectors, each with six-fold segmentation and arranged in eight triple-clusters. The scattered projectiles and target recoils were detected in a highly segmented silicon detector[19]. See Methods.

Prior to the present work, nothing was known about the energies and spins of excited states in $^{224,226}$Rn, while de-exciting γ-rays from states in $^{222}$Rn had been observed[20] with certainty up to $I^\pi = 13^-$. The chosen bombarding energies for $^{224,226}$Rn were about 3% below the nominal Coulomb barrier energy at which the beam and target nuclei come close enough in head-on collisions for nuclear forces to significantly influence the reaction mechanism. For such close collisions the population of high-spin states will be enhanced, allowing the rotational behaviour of the nucleus to be elucidated. This is the method described by Ward et al.[21] and has subsequently been coined unsafe Coulomb excitation[22] as the interactions between the high-Z reaction partners is predominantly electromagnetic. It is not possible to precisely determine electromagnetic matrix elements because of the small nuclear contribution. The most intense excited states expected to be observed belong to the positive-parity rotational band, built upon the ground state. These states are connected by fast electric quadrupole (E2) transitions. In nuclei that are unstable to pear-shaped distortion, the other favoured excitation paths are to members of the octupole band, negative-parity states connected to the ground-state band by strong E3 transitions.

The spectra of γ-rays time-correlated with scattered beam and target recoils are shown in Fig. 1. The E2 γ-ray transitions within the ground-state positive-parity band can be clearly identified, as these de-excite via a regular sequence of strongly-excited states having spin and parity $0^+, 2^+, 4^+, \ldots$ with energies $\frac{\hbar^2}{2\Im} I(I+1)$. In this expression the moment-of-inertia $\Im$ systematically increases with increasing $I$ (reducing pairing) and with number of valence nucleons (increasing quadrupole deformation). As expected from multi-step Coulomb excitation the intensities of the transitions systematically decrease with increasing $I$, after correcting for internal conversion and the γ-ray detection efficiency of the Miniball array.

The other relatively intense γ-rays observed in these spectra with energies <600 keV are assumed to have electric-dipole (E1) multipolarity, and to depopulate the odd-spin negative-parity members of the octupole band. In order to determine which states are connected by these transitions, pairs of time-correlated ('coincident') γ-rays were examined. In this analysis, the energy spectrum of γ-rays coincident with one particular transition is generated by requiring that the energy of this 'gating' transition lies in a specific range. Typical spectra obtained this way are shown in Fig. 2. Each spectrum corresponds to a particular gating transition, background subtracted, so that the peaks observed in the spectrum arise from γ-ray transitions in coincidence with that transition.

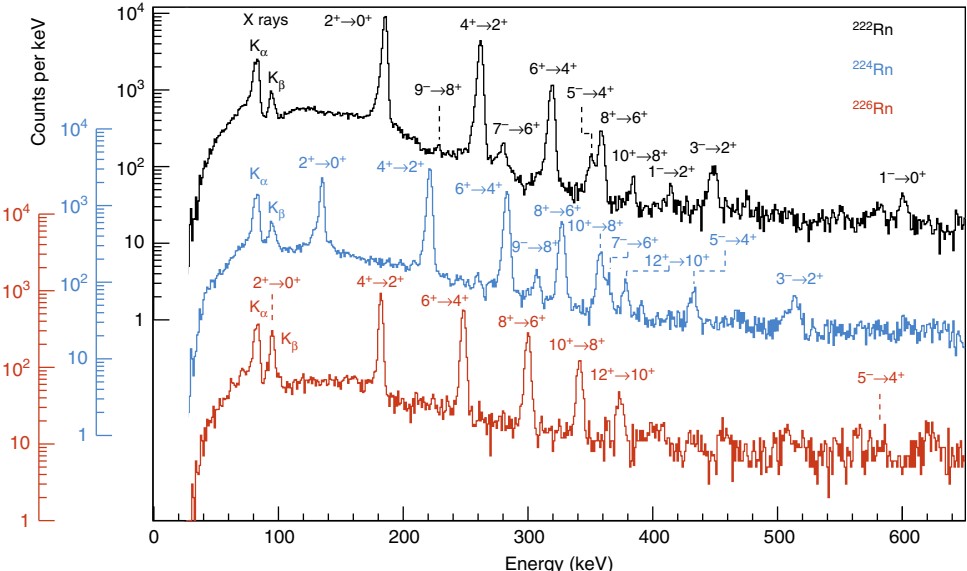

**Fig. 1** Spectra of γ-rays. The γ-rays were emitted following the bombardment of $^{120}$Sn targets by $^{222}$Rn (black), $^{224}$Rn (blue) and $^{226}$Rn (red). The γ-rays were corrected for Doppler shift assuming that they are emitted from the scattered projectile. Random coincidences between Miniball and CD detectors have been subtracted. The transitions which give rise to the observed full-energy peaks are labelled by the spin and parity of the initial and final quantum states. The assignments of the transitions from the negative-parity states in $^{224,226}$Rn are tentative (see text)

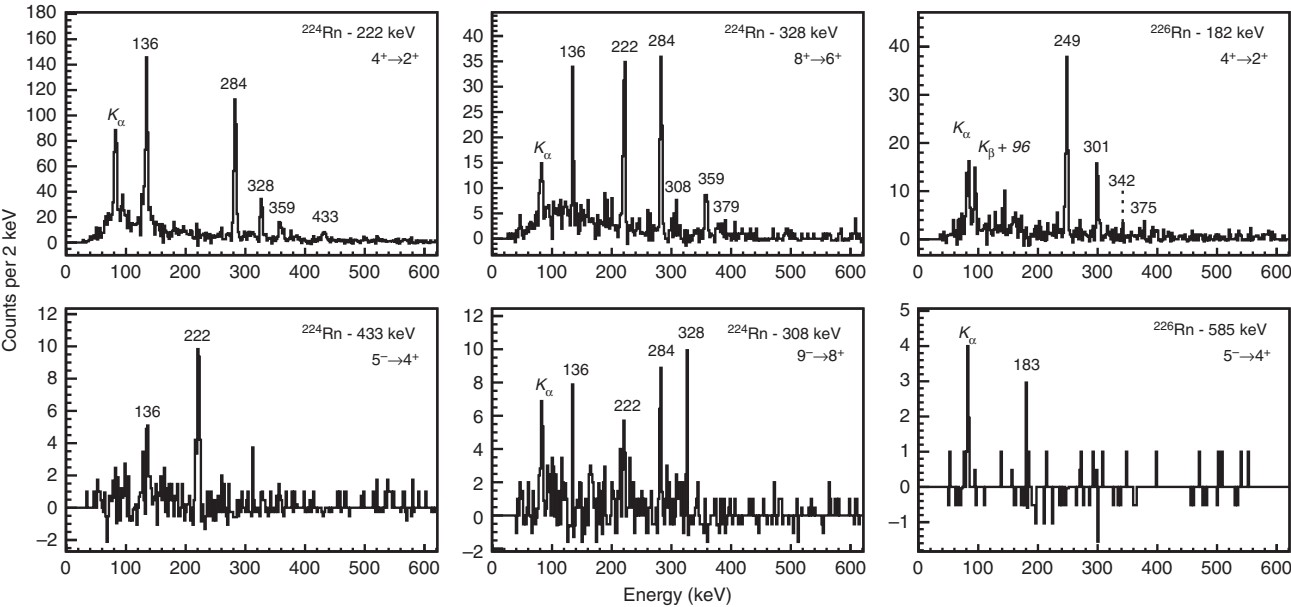

**Fig. 2** Coincidence γ-ray spectra. The representative background-subtracted γ-ray spectra are in time-coincidence with different gating transitions. Here the observed peaks are labelled by the energy (in keV) of the transition. The gating transition is additionally labelled by the proposed spin and parity of the initial and final states

The level schemes for $^{224,226}$Rn constructed from the coincidence spectra, together with the known[20] scheme for $^{222}$Rn, are shown in Fig. 3. For $^{226}$Rn the energy of the strongly-converted $2^+ \rightarrow 0^+$ transition overlaps with those of the $K_\beta$ X-rays, but its value can be determined assuming that the relative intensity of $K_\beta$, $K_\alpha$ X-rays is the same as for $^{222,224}$Rn. The E2 transitions connecting the states in the octupole band are not observed because they cannot compete with faster, higher-energy E1 decays. The only other plausible description for this band is that it has $K^\pi = 0^+$ or $2^+$, implying that the $K^\pi = 0^-$ octupole band is not observed. (Here $K$ is the projection of $\boldsymbol{I}$ on the body-fixed symmetry axis.) This is unlikely as the bandhead would have

to lie significantly lower in energy than has been observed in $^{222,224,226}$Ra, and inter-band transitions from states with $I' > 4$ to states with $I$ and $I$-2 in the ground-state band and in-band transitions to $I'$-2 would all be visible in the spectra. The spin and parity assignments for the positive-parity band that is strongly populated by Coulomb excitation can be regarded as firm, whereas the negative-parity state assignments are made in accord with the systematic behaviour of nuclei in this mass region.

**Characterisation of octupole instability.** From the level schemes and from the systematics for all the radon isotopes (Fig. 4) it is

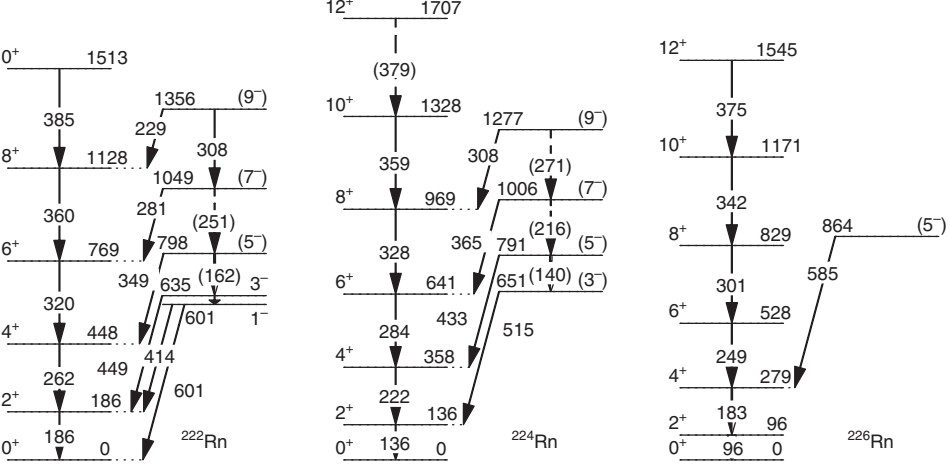

**Fig. 3** Level schemes. These partial level schemes for $^{222,224,226}$Rn show the excited states of interest. Arrows indicate γ-ray transitions. All energies are in keV. Firm placements of transitions in the scheme are from previous work[20] or have been made using γ-γ-coincidence relations; otherwise in brackets

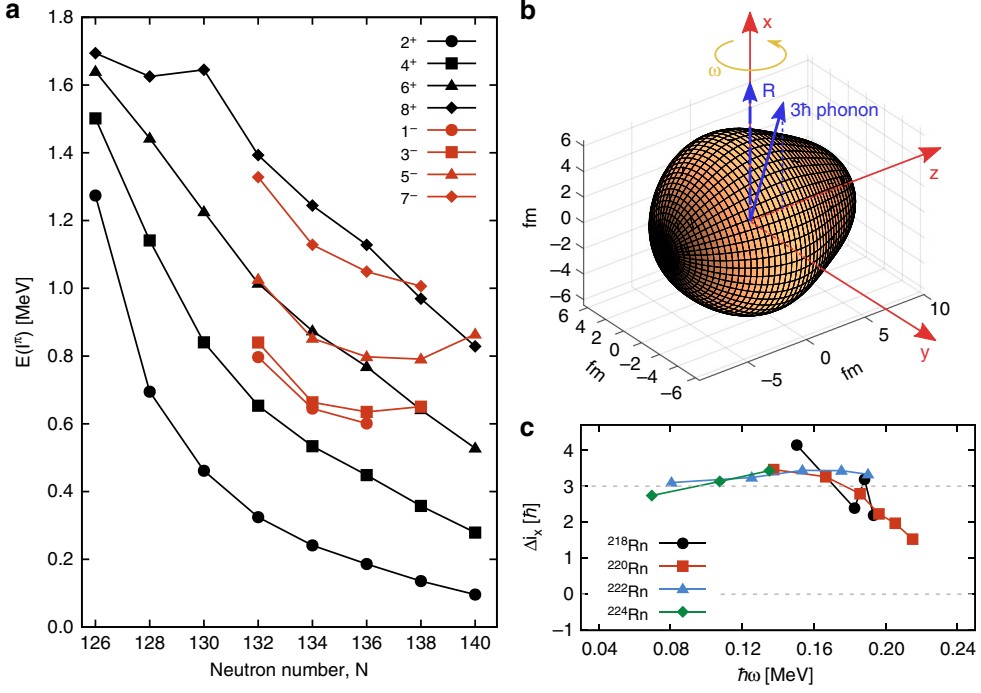

**Fig. 4** Systematic behaviour of radon isotopes. **a** Systematics of the energies for different spins of low-lying positive-parity (black) and negative-parity states (red) in radon isotopes; **b** cartoon illustrating how the octupole phonon vector aligns with the rotation ($R$) vector (which is orthogonal to the rotating body's symmetry axis) so that $I = R + 3\hbar$ and $\Delta i_x = 3\hbar$; **c** difference in aligned spin for negative- and positive-parity states in $^{218-224}$Rn (re-analysed for $^{218-222}$Rn that have been presented earlier[23]). The dashed line at $\Delta i_x = 0$ is the expected value for static-octupole deformation

clear that the bandhead of the octupole band reaches a minimum around $N = 136$. The character of the octupole bands can be explored[23] by examining the difference in aligned angular momentum, $\Delta i_x = i_x^- - i_x^+$, at the same rotational frequency $\omega$, as a function of $\omega$. Here $i_x$ is approximately $I$ for $K=0$ bands and $\hbar\omega$ is approximately $(E_I - E_{I-2})/2$. For nuclei with permanent octupole deformation $\Delta i_x$ is expected to approach zero, as observed for several isotopes of Ra, Th, and U[5]. For octupole-vibrational nuclei in which the negative-parity states arise from coupling an octupole phonon to the positive-parity states, it is expected that $\Delta i_x \sim 3\hbar$ as the phonon prefers to align with the rotational axis. This is the case for the isotopes $^{218,220,222,224}$Rn at values of $\hbar\omega$ (<0.2 MeV) where particle-hole excitations do not play a role, see Fig. 4. Thus we have clearly delineated the lower

boundary at $Z > 86$ as to where permanent octupole deformation occurs in nature.

## Discussion

The observation of octupole-vibrational bands in the even-even radon isotopes is consistent with several theoretical calculations[24–26], which predict that only nuclei with $Z > 86$ have stable octupole deformation. Other calculations suggest that radon isotopes with A~222 will have non-zero values of the octupole deformation parameter $\beta_3$[27,28]. For such nuclei, which have a minimum in the nuclear potential energy at non-zero values of $\beta_3$, the positive- and negative-parity states are projected from intrinsic configurations having $K^\pi = 0^+, 0^-$, which are degenerate in energy. In the odd-A neighbours parity doublets arise by

**Table 1 Energies of levels and transitions in $^{224}$Rn and $^{226}$Rn**

| $^{224}$Rn $E_{level}$ (keV) | $I_i^\pi$ | $E_\gamma$ (keV) | $I_f^\pi$ |
|---|---|---|---|
| 135.6 (5) | $2^+$ | 135.6 (5) | $0^+$ |
| 357.6 (6) | $4^+$ | 222.0 (5) | $2^+$ |
| 641.4 (8) | $6^+$ | 283.8 (5) | $4^+$ |
| 650.6 (8) | $(3^-)$ | 515.0 (6) | $2^+$ |
| 790.8 (8) | $(5^-)$ | 433.2 (5) | $4^+$ |
| 969.2 (9) | $8^+$ | 327.8 (5) | $6^+$ |
| 1006.4 (10) | $(7^-)$ | 365.0 (5) | $6^+$ |
| 1277.2 (10) | $(9^-)$ | 308.0 (5) | $8^+$ |
| 1327.8 (10) | $10^+$ | 358.6 (5) | $8^+$ |
| 1706.8 (11) | $(12^+)$ | 379.1 (5) | $10^+$ |

| $^{226}$Rn $E_{level}$ (keV) | $I_i^\pi$ | $E_\gamma$ (keV) | $I_f^\pi$ |
|---|---|---|---|
| 96.0 (11) | $2^+$ | 96.0 (11) | $0^+$ |
| 278.9 (12) | $4^+$ | 182.9 (5) | $2^+$ |
| 527.9 (13) | $6^+$ | 249.0 (5) | $4^+$ |
| 828.6 (14) | $8^+$ | 300.7 (5) | $6^+$ |
| 864 (2) | $(5^-)$ | 585.4 (18) | $4^+$ |
| 1170.8 (14) | $10^+$ | 342.1 (5) | $8^+$ |
| 1545.4 (15) | $12^+$ | 374.6 (5) | $10^+$ |

The 1-σ errors are given, estimated from the statistical error and the uncertainty in the energy calibration and Doppler correction

to a scattering angular range of 140.9°–0.2° in the centre-of-mass. In order to reduce background from Compton scattering, events were rejected if any two germanium crystals in each triple-cluster registered simultaneous γ-ray hits, in contrast to the normal adding procedure which would substantially increase the probability of summing two γ-rays emitted in the same decay sequence ('true pile-up'). Miniball was calibrated using $^{133}$Ba and $^{152}$Eu radioactive sources that emitted γ-rays of known energy and relative intensity. The relativistic Doppler correction was performed by determining the momentum vector of the projectile, using the energy and position information in the pixelated silicon detector, and the emission polar and azimuthal angle of the detected gamma-ray in the segment of Miniball where most energy was deposited. In the case of the latter the relative orientation of each segment to each other and to the beam axis was determined by employing d($^{22}$Ne,pγ) and d($^{22}$Ne,nγ) reactions. The Doppler corrected energies for transitions in $^{224}$Rn and $^{226}$Rn together with the deduced level energies are given in Table 1.

## Data availability

The data that support the findings of this study are available from the corresponding author on reasonable request. The software used to sort the raw data is available at https://doi.org/10.5281/zenodo.2593370 (Gaffney, L. P. & Konki, J., MiniballCoulexSort for Coulex, SPEDE, CREX and TREX). Information about the ROOT software package used to analyse the data can be found at https://doi.org/10.1016/j.cpc.2009.08.005.

coupling the odd particle to these configurations. This is not the case for reflection-symmetric nuclei that undergo octupole vibrations around $\beta_3 = 0$. Bands of opposite parity with differing single-particle configurations can lie close to each other fortuitously[29,30] but in general those arising from coupling the odd nucleon to the ground state and octupole phonon will be well separated. The separation will be determined by the spacing of the bands in the even-even core, ~500 keV in the case of $^{222-226}$Rn (see Fig. 4), and will be in general much larger than that the value (~50 keV) observed for parity doublets in radium isotopes[1]. Quantitative estimates of Schiff moments for octupole-vibrational systems have yet to be made[31]. Nevertheless, it can be concluded that, if measurable CP-violating effects occur in nuclei, the enhancement of nuclear Schiff moments arising from octupole effects in odd-$A$ radon nuclei is likely to be much smaller than for heavier octupole-deformed systems.

## Methods

**Production of radioactive radon beams.** In our experiments, $^{222,224,226}$Rn were produced by spallation in the primary target, diffused to the surface and then singly ionized ($q = 1^+$) in an enhanced plasma ion-source[32] with a cooled transfer line. The ions were then accelerated to 30 keV, separated according to $A/q$, and delivered to a Penning trap, REXTRAP[33], at a rate of around $8 \times 10^6$ ions s$^{-1}$ for $^{222}$Rn, $2 \times 10^6$ ions s$^{-1}$ for $^{224}$Rn and $10^5$ ions s$^{-1}$ for $^{226}$Rn at the entrance. Inside the trap, the singly-charged ions were accumulated and cooled before being allowed to escape in bunches at 500 ms intervals into an electron-beam ion source, REXE-BIS[33]. Here, the ions were confined for 500–700 ms in a high-density electron beam that stripped more electrons to produce a charge state of $51^+$ ($^{222}$Rn) or $52^+$ ($^{224,226}$Rn) extracted as 1 ms pulses before being mass-selected again according to $A/q$, and injected at 2 Hz into the HIE-ISOLDE linear post-accelerator. The Rn beams then bombarded a $^{120}$Sn target of thickness ~2 mg cm$^{-2}$ with an intensity of about $6 \times 10^5$ ions s$^{-1}$, $1.1 \times 10^5$ ions s$^{-1}$ and $2 \times 10^3$ ions s$^{-1}$ for $^{222}$Rn, $^{224}$Rn and $^{226}$Rn, respectively. The total beam-times were respectively 8, 16 and 24 h. The level of Fr impurity in the Rn beams could be estimated for $A = 222$ as below 1% by observing radioactive decays at the end of the beam line.

**Data selection and Doppler correction.** Events corresponding to the simultaneous detection of γ-rays and heavy ions in Miniball and the silicon detector array respectively were selected if the measured energy and angle of either projectile or target satisfied the expected kinematic relationship for inelastic scattering reactions. This procedure eliminated any background from stable noble-gas contaminant beams produced in REX-TRAP having the same $A/q$ as the radon beams. In the present setup the average angle of each of the 16 strips of one side of the silicon detector array ranged between 19.6° and 54.9° to the beam direction, corresponding

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

## Acknowledgements

We are grateful to Niels Bidault, Eleftherios Fadakis, Erwin Siesling, and Fredrick Wenander who assisted with the preparation of the radioactive beams, and we thank Jacek Dobaczewski for useful discussions. The support of the ISOLDE Collaboration and technical teams is acknowledged. This work was supported by the following Research Councils and Grants: Science and Technology Facilities Council (STFC; UK) grants ST/P004598/1, ST/L005808/1; Federal Ministry of Education and Research (BMBF; Germany) grants 05P18RDCIA, 05P15PKCIA and 05P18PKCIA and the "Verbundprojekt 05P2018"; National Science Centre (Poland) grant 2015/18/M/ST2/00523; European Union's Horizon 2020 Framework research and innovation programme 654002 (ENSAR2); Marie Skłodowska-Curie COFUND grant (EU-CERN) 665779; Research Foundation Flanders (FWO, Belgium), by GOA/2015/010 (BOF KU Leuven) and the Interuniversity Attraction Poles Programme initiated by the Belgian Science Policy Office (BriX network P7/12); RFBR(Russia) grant 17-52-12015.

## Author contributions

P.A.B., T.C., L.P.G., M.Sc., N.W. and M.Z. prepared the proposal for the experiment, K.A., H.D.W., L.P.G., A.I., J.K., P.R., D.R., M.Se., V.V., and N.W. set up the instrumentation, L.P.G., K.J., M.L., J.A.R., and S.R. prepared the radioactive beams, M.B., P.A.B., J.C., G.D.A., L.P.G., P.E.G., A.G., C.H., D.T.J., J.M.K., N.A.K., M.K., J.K., T.K., B.N.S., D.O.D., J.O., R.D.P., L.G.P., C.R., M.Sc., T.S., B.S., J.Si., J.F.S., P.S., M.St., S.V., V.V., N.W., K.W.L., M.Z. monitored the detector, data acquisition and radioactive beam systems, P.A.B., L.P.G., J.K., J.F.S., P.S., P.V.D., and N.W. carried out the data analysis and interpretation of the data, and P.A.B., T.C., G.D.A., L.P.G., J.K., T.K., B.N.S., J.O., M.Sc., J.F.S., P.S., P.V.D., N.W., and K.W.L. prepared the manuscript.

## Additional information

**Competing interests:** The authors declare no competing interests.

