## [Peer Review File · Nature Communications]

Reviewers' comments:

Reviewer #1 (Remarks to the Author):

The investigation of static octupole moments in the intrinsic frame of atomic nuclei has gained a strong interest in recent years. Such intrinsic moments would increase considerably the possibility to observe nonvanishing electric dipole moments of particles, atoms, or molecules with a nondegenerate ground state. This could give a hint for the violation of basic symmetries and for physics beyond the standard model.

Since the early days of nuclear physics, it is well known that there exist in the nuclear chart a large number of nuclei with intrinsic static quadrupole deformations having reflection symmetry. However, there are only a few regions with intrinsic static octupole deformation. The best-known cases are nuclei in the Radium and Thorium region with a charge number $Z > 86$. It is therefore of great interest to search for such cases also in lighter Actinides.

The present authors use two heavy even-even Radon isotopes ^{224}Ra and ^{226}Ra and search for experimental evidence of facts connected with intrinsic static octupole shapes and with the violation of the intrinsic reflection symmetry. These rare isotopes are produced by spallation of a Thorium target bombarded with high energy protons from the CERN PS Booster, afterward accelerated in HIE-ISOLDE and Coulomb excited by a secondary ^{120}Sn target. The emitted γ -decay and the corresponding spectra are analyzed critically in detail. As a result, the authors find no low-lying parity doublets indicating intrinsic static octupole shapes in these two isotopes. They only find negative parity bands with band heads at excitation energy above 500 keV which fit well into the systematic of octupole vibrations in lighter Ra isotopes known from earlier experiments.

Therefore the authors conclude that in these two isotopes there are no favorable conditions, which could increase the possibility to measure atomic electric dipole moments in this region.

These experimental results are also supported by a number of theoretical investigations (Refs. [15-17]) based on modern state of the art energy density functional theories, which provide a rather successful and universal description of shapes of nuclei all over the periodic table and which indicate clearly that in the Actinides static octupole deformations are only found above $Z > 86$.

The paper is very clearly written and easily understandable for a general readership not specialized in the details of γ -spectroscopy. It contains important results for future experiments concerned with the very hot topic violation of basic symmetries and the physics beyond the standard model. Therefore I fully support the publication of this manuscript in Nature Communications.

Reviewer #2 (Remarks to the Author):

Re: NCOMMS-19-06076-T

The article "The observation of vibrating pear-shapes in radon nuclei" by P.A. Butler et al., provides the energies and spins of excited states in $^{224,226}\text{Rn}$ for the first time, which reveals that these isotopes are not good candidates for static octupole deformation. This conclusion is supported by theoretical calculations [15-17]. Therefore, these isotopes are not favorable for enhancing a possible atomic electric-dipole moment. This is a very important result considering the interest in the Rn and Ra isotopes for atomic EDM measurements. This work builds off of recent work on ^{220}Rn and ^{224}Ra by the same authors [Nature 497, 199 (2013)], which reported favorable conditions for ^{224}Ra .

I have two primary criticisms that need to be addressed before I can make a recommendation:

(1) the authors do not report any matrix elements, which were given in their Nature article for ^{220}Rn and ^{222}Ra . This makes the story seem incomplete. Considering the authors take the time to state,

"The chosen bombarding energies for $^{224,226}\text{Rn}$ were about 3% below the nominal "Coulomb barrier energy" at which the beam and target nuclei come close enough in head-on collisions for nuclear forces to influence the reaction mechanism. At these energies the two-body reactions selected by the silicon detector with scattering angle (centre-of-mass) $< 140^\circ$ are predominantly electromagnetic (Coulomb excitation).",

I was expecting matrix elements. If the authors only intend to discuss energies, then the degree to which there was "safe Coulex" seems irrelevant. What bearing does the selection of $\theta_{\text{cm}} < 140$ degrees have on reporting energies?

I also disagree on the statement that the system is "safe" for < 140 degrees with a $^{224,226}\text{Rn}$ beam energy of 5.08 MeV per nucleon on a ^{120}Sn target. The nominal "safe" energy is 4.22 MeV per nucleon and the nominal safe distance is 18.76 fm. At 5.08 MeV per nucleon, the safe distance isn't satisfied until $\theta_{\text{cm}} < 90$ degrees. At the center of the ^{120}Sn target, safe angles are for < 105 degrees.

My suspicion is that the states of interest are only seen when unsafe angles are included. This would be fine if only energies are reported, as done in the present paper, but the authors shouldn't imply that more is possible without additional clarification. Otherwise, I recommend that the authors remove all statements concerning "safe" Coulex.

(2) the authors do not properly communicate the specific interest in $^{222,224,226}\text{Rn}$ for atomic EDM searches or point towards any expectations that these isotopes would have static octupole deformation. The theoretical papers mentioned [15-17] did not expect or predict static octupole deformation. If one digs into Refs. [2-4], specific interest in ^{223}Rn can be found, where $^{222,224}\text{Rn}$ are relevant. Has there been any serious interest in ^{225}Rn , particularly considering the difficulty in producing these nuclei? I think more explanation towards the interest of these specific isotopes is required, particularly for a reader that is not an expert.

Reviewer #3 (Remarks to the Author):

I have read the manuscript entitled "The observation of vibrating pear-shapes in radon nuclei" by Prof Butler and colleagues, which was submitted for publication in Nature Communications. The manuscript reports on an excellent piece of work by a world-class team.

It is a strong piece of work. It is, however, hard to find the stand out achievement (technical, conceptual, intellectual, or otherwise) that would warrant publication in a journal of this rank.

The abstract and introduction are elegant and concise that I see no sense in regurgitating a lengthy summary of what is perfectly well done by these experienced authors on page one of the manuscript. Briefly: a static octupole deformation in nuclei in particular region of the nuclear chart would make them significantly beneficial to the searches for permanent electric-dipole moments (EDMs) larger in magnitude than the standard model predicts. Experiments, as presented here, can probe such deformation. A subset of these authors has demonstrated this emphatically in a 2013 work in published in Nature, concerning the Ra isotopes. The Rn isotopes, reported on here, do not, as suspected, present such a case. Were a checklist of such things in existence, Rn has now been checked off.

The subject of the paper is highly topical and, along with the authors own works, led to a recent Nature paper (Gaffney et al. 2013) and Physical Review Letters (Bucher et al. 2016, 2017) and so on. Each of those works represented, to various degrees, milestone measurements.

The work as a whole is concise, simple, presents its intent in a clear and defined manner, a clear and simple description of the analysis and discussion and a simple conclusion, in line with what theory had suggested. It is carried out by a world-class team at a world-class facility, using state-of-the-art equipment. It uses references appropriately. I have no concerns about what is presented on any level and it is quite difficult to find fault with the work. In short, it is a solid piece of work.

In part quoting the recommendations laid forth by the journal, I can acknowledge that the data are technically sound, the paper provides strong evidence for its conclusions, and the manuscript is important to scientists in the specific field.

However, the results are objectively not particularly novel, nor likely represent an advance in understanding likely to influence thinking in the field. Much of the critical thinking has been done and presented in the works referenced from the 90s, 00s, and beyond, only now is the community able to access some of these systems experimentally. To meet the high standards of Nature (Communications) it does not demonstrate novelty, being a somewhat routine Coulomb excitation measurement of the likes the same group and facility has done before. Neither is there a major conceptual advance. These nuclei have been listed or targeted for many decades as worthy of consideration or to be ruled out; it is somewhat of a checklist item. As for major technical innovations, the important aspect here was the facility HIE-ISOLDE (major results already seen in a Physical Review Letters paper on Coulomb excitation on Sn), but again, it compares to other works preceding it. In terms of interpretational problems, it is a conventional interpretation of spectra, using common analysis techniques established over decades. I have attempted to be objective in these remarks.

The authors somewhat point out what would be the standout measurement or result that would be suitable for Nature (Comms) by stating "Direct observation of low-lying states in odd-A Rn nuclei (e.g. following Coulomb excitation or alpha-decay from the astatine parent) is presently not possible, as it will require significant advances in the technology used to produce radioactive ions."

Prof Butler and colleagues should be commended on their excellent, incremental work. It is ripe for publication in a dedicated journal on nuclear physics and its applications to fundamental symmetries.

Reviewer #1 (Remarks to the Author):

The investigation of static octupole moments in the intrinsic frame of atomic nuclei has gained a strong interest in recent years. Such intrinsic moments would increase considerably the possibility to observe nonvanishing electric dipole moments of particles, atoms, or molecules with a nondegenerate ground state. This could give a hint for the violation of basic symmetries and for physics beyond the standard model.

Since the early days of nuclear physics, it is well known that there exist in the nuclear chart a large number of nuclei with intrinsic static quadrupole deformations having reflection symmetry. However, there are only a few regions with intrinsic static octupole deformation. The best-known cases are nuclei in the Radium and Thorium region with a charge number $Z > 86$. It is therefore of great interest to search for such cases also in lighter Actinides.

The present authors use two heavy even-even Radon isotopes ^{224}Ra and ^{226}Ra and search for experimental evidence of facts connected with intrinsic static octupole shapes and with the violation of the intrinsic reflection symmetry. These rare isotopes are produced by spallation of a Thorium target bombarded with high energy protons from the CERN PS Booster, afterward accelerated in HIE-ISOLDE and Coulomb excited by a secondary ^{120}Sn target. The emitted γ -decay and the corresponding spectra are analyzed critically in detail. As a result, the authors find no low-lying parity doublets indicating intrinsic static octupole shapes in these two isotopes. They only find negative parity bands with band heads at excitation energy above 500 keV which fit well into the systematic of octupole vibrations in lighter Ra isotopes known from earlier experiments.

Therefore the authors conclude that in these two isotopes there are no favorable conditions, which could increase the possibility to measure atomic electric dipole moments in this region.

These experimental results are also supported by a number of theoretical investigations (Refs. [15-17]) based on modern state of the art energy density functional theories, which provide a rather successful and universal description of shapes of nuclei all over the periodic table and which indicate clearly that in the Actinides static octupole deformations are only found above $Z > 86$.

The paper is very clearly written and easily understandable for a general readership not specialized in the details of γ -spectroscopy. It contains important results for future experiments concerned with the very hot topic violation of basic symmetries and the physics beyond the standard model. Therefore I fully support the publication of this manuscript in Nature Communications.

Reviewer #2 (Remarks to the Author):

Re: NCOMMS-19-06076-T

The article "The observation of vibrating pear-shapes in radon nuclei" by P.A. Butler et al., provides the energies and spins of excited states in $^{224,226}\text{Rn}$ for the first time, which reveals that these isotopes are not good candidates for static octupole deformation. This conclusion is supported by theoretical calculations [15-17]. Therefore, these isotopes are not favorable for enhancing a possible atomic electric-dipole moment. This is a very important result considering the interest in the Rn and Ra isotopes for atomic EDM measurements. This work builds off of recent work on ^{220}Rn and ^{224}Ra by the same authors [Nature 497, 199 (2013)], which reported favorable conditions for ^{224}Ra .

I have two primary criticisms that need to be addressed before I can make a recommendation:

(1) the authors do not report any matrix elements, which were given in their Nature article for ^{220}Rn and ^{224}Ra . This makes the story seem incomplete. Considering the authors take the time to state,

"The chosen bombarding energies for $^{224,226}\text{Rn}$ were about 3% below the nominal "Coulomb barrier energy" at which the beam and target nuclei come close enough in head-on collisions for nuclear forces to influence the reaction mechanism. At these energies the two-body reactions selected by the silicon detector with scattering angle (centre-of-mass) $< 140^\circ$ are predominantly electromagnetic (Coulomb excitation).",

I was expecting matrix elements. If the authors only intend to discuss energies, then the degree to which there was "safe Coulex" seems irrelevant. What bearing does the selection of $\theta_{\text{cm}} < 140$ degrees have on reporting energies?

I also disagree on the statement that the system is "safe" for < 140 degrees with a $^{224,226}\text{Rn}$ beam energy of 5.08 MeV per nucleon on a ^{120}Sn target. The nominal "safe" energy is 4.22 MeV per nucleon and the nominal safe distance is 18.76 fm. At 5.08 MeV per nucleon, the safe distance isn't satisfied until $\theta_{\text{cm}} < 90$ degrees. At the center of the ^{120}Sn target, safe angles are for < 105 degrees.

My suspicion is that the states of interest are only seen when unsafe angles are included. This would be fine if only energies are reported, as done in the present paper, but the authors shouldn't imply that more is possible without additional clarification. Otherwise, I recommend that the authors remove all statements concerning "safe" Coulex.

We thank the referee for highlighting omissions in the discussion of the experimental method that was employed. We tried not to imply that we are employing "safe" Coulex that enables electromagnetic matrix elements to be determined; instead we asserted that under the conditions of bombarding energy and scattering angles employed the excitation mechanism for these high Z projectiles and target is mostly electromagnetic in nature. In fact we are using the method of "unsafe Coulex" (as the referee has pointed out) in order to enhance the population of high spin

states, necessary to elucidate the decay scheme. We have clarified the text and have now specifically referred to this technique. The discussion of the details of the scattering angle acceptance has been moved to the methods summary.

(2) the authors do not properly communicate the specific interest in $^{222,224,226}\text{Rn}$ for atomic EDM searches or point towards any expectations that these isotopes would have static octupole deformation. The theoretical papers mentioned [15-17] did not expect or predict static octupole deformation. If one digs into Refs. [2-4], specific interest in ^{223}Rn can be found, where $^{222,224}\text{Rn}$ are relevant. Has there been any serious interest in ^{225}Rn , particularly considering the difficulty in producing these nuclei? I think more explanation towards the interest of these specific isotopes is required, particularly for a reader that is not an expert.

We have expanded the text to include additional references that contain theoretical predictions for stable octupole deformation in the radon isotopes, although these publications do not provide the same level of detail as the ones that we originally chose. We have also expanded the discussion, including additional references that demonstrate the interest in searching for EDMs in radon atoms. Note that the expected production rate of ^{225}Rn from HIE-ISOLDE is one order of magnitude lower than that of ^{223}Rn , but only a factor of 3 lower for the future FRIB facility.

Reviewer #3 (Remarks to the Author):

I have read the manuscript entitled "The observation of vibrating pear-shapes in radon nuclei" by Prof Butler and colleagues, which was submitted for publication in Nature Communications. The manuscript reports on an excellent piece of work by a world-class team.

It is a strong piece of work. It is, however, hard to find the stand out achievement (technical, conceptual, intellectual, or otherwise) that would warrant publication in a journal of this rank.

The abstract and introduction are elegant and concise that I see no sense in regurgitating a lengthy summary of what is perfectly well done by these experienced authors on page one of the manuscript. Briefly: a static octupole deformation in nuclei in particular region of the nuclear chart would make them significantly beneficial to the searches for permanent electric-dipole moments (EDMs) larger in magnitude than the standard model predicts. Experiments, as presented here, can probe such deformation. A subset of these authors has demonstrated this emphatically in a 2013 work in published in Nature, concerning the Ra isotopes. The Rn isotopes, reported on here, do not, as suspected, present such a case. Were a checklist of such things in existence, Rn has now been checked off.

We disagree with the statement that Rn has been "checked off" as a candidate for static octupole deformation and EDM candidate from our earlier 2013 work published in Nature. In fact this work measured the properties of Rn-220 and did not make any observations about heavier radon (222, 224, 226) isotopes, which could well develop pear-shapes. The earlier work is therefore not relevant to EDM searches in the heavier radon atoms where parity doublets are more likely to occur. We have clearly delineated the lower boundary where permanent octupole deformation occurs in nature and hence where parity doublets might be expected to occur. We have clarified this in additional text that also provides additional references to prospective radon EDM searches. The abstract has also been slightly modified so that it is consistent with the last paragraph that summarizes implications for EDM searches.

The subject of the paper is highly topical and, along with the authors own works, led to a recent Nature paper (Gaffney et al. 2013) and Physical Review Letters (Bucher et al. 2016, 2017) and so on. Each of those works represented, to various degrees, milestone measurements.

The recent publications by Bucher et al. (2016,2017) report measurements of B(E3) matrix elements in 144,146Ba. The barium atoms have much lower Z than radon and therefore are not expected to offer the same Schiff moment enhancement, even though they have strong octupole correlations. In addition, parity doublets have not been observed in odd-mass barium isotopes, a necessary condition for Schiff enhancement. Atoms in this mass region have therefore not been identified as candidates for EDM searches so these publications are not particularly relevant to the present work.

The work as a whole is concise, simple, presents its intent in a clear and defined manner, a clear and simple description of the analysis and discussion and a simple conclusion, in line with what theory

had suggested. It is carried out by a world-class team at a world-class facility, using state-of-the-art equipment. It uses references appropriately. I have no concerns about what is presented on any level and it is quite difficult to find fault with the work. In short, it is a solid piece of work.

In part quoting the recommendations laid forth by the journal, I can acknowledge that the data are technically sound, the paper provides strong evidence for its conclusions, and the manuscript is important to scientists in the specific field.

However, the results are objectively not particularly novel, nor likely represent an advance in understanding likely to influence thinking in the field. Much of the critical thinking has been done and presented in the works referenced from the 90s, 00s, and beyond, only now is the community able to access some of these systems experimentally. To meet the high standards of Nature (Communications) it does not demonstrate novelty, being a somewhat routine Coulomb excitation measurement of the likes the same group and facility has done before. Neither is there a major conceptual advance. These nuclei have been listed or targeted for many decades as worthy of consideration or to be ruled out; it is somewhat of a checklist item. As for major technical innovations, the important aspect here was the facility HIE-ISOLDE (major results already seen in a Physical Review Letters paper on Coulomb excitation on Sn), but again, it compares to other works preceding it. In terms of interpretational problems, it is a conventional interpretation of spectra, using common analysis techniques established over decades. I have attempted to be objective in these remarks.

We respectfully point out that there is nothing routine about accelerating rare, radioactive isotopes of the very heavy nuclei used in the present measurements. HIE-ISOLDE is the only facility, world-wide, capable of doing this and has only recently been able to accelerate heavy ions such as radon to sufficiently high energy to enable the observations reported here. This has followed decades of technical development at CERN. Of course we are using established analysis techniques, but this paper presents the first application of these techniques to exotic, radioactive beams with intensities 5 orders of magnitude weaker than for conventional stable beams.

The authors somewhat point out what would be the standout measurement or result that would be suitable for Nature (Comms) by stating “Direct observation of low-lying states in odd-A Rn nuclei (e.g. following Coulomb excitation or alpha-decay from the astatine parent) is presently not possible, as it will require significant advances in the technology used to produce radioactive ions.”

Prof Butler and colleagues should be commended on their excellent, incremental work. It is ripe for publication in a dedicated journal on nuclear physics and its applications to fundamental symmetries.

The topics of nuclear physics and fundamental symmetries are quite different and such a dedicated journal does not exist. We believe that a paper covering topics of interest to the nuclear physics and particle physics community, and judging by the response to our earlier 2013 work a much wider scientific community (84th percentile of all Nature articles of a similar age), is best published in Nature Communications.

REVIEWERS' COMMENTS:

Reviewer #2 (Remarks to the Author):

Re: NCOMMS-19-06076-T

The authors have addressed my concerns (and those of the other referees) to my satisfaction and I am now happy to accept this paper for publication.

Reviewer #3 (Remarks to the Author):

This reviewer provided confidential remarks only to the editors.